# Arbuscular Mycorrhizal Fungal Assemblages Significantly Shifted upon Bacterial Inoculation in Non-Contaminated and Petroleum-Contaminated Environments

**DOI:** 10.3390/microorganisms8040602

**Published:** 2020-04-21

**Authors:** Dimitri J. Dagher, Ivan E. de la Providencia, Frédéric E. Pitre, Marc St-Arnaud, Mohamed Hijri

**Affiliations:** 1Institut de Recherche en Biologie Végétale, Université de Montréal and Jardin botanique de Montréal, 4101 Sherbrooke est, Montréal, QC H1X 2B2, Canada; dimitri.dagher@umontreal.ca (D.J.D.); frederic.pitre@umontreal.ca (F.E.P.); marc.st-arnaud@umontreal.ca (M.S.-A.); 2Neomed-labs 525 Boulevard Cartier O, Laval, QC H7V 4A2, Canada; iprovidencia@yahoo.com; 3AgroBioSciences, University Mohammed VI Polytechnic, Lot 660–Hay Moulay Rachid, Ben Guerir 43150, Morocco

**Keywords:** Arbuscular mycorrhizal fungi, plant–microbes interactions, microbial ecology, rhizosphere microbiome, bacterial inoculation, petroleum hydrocarbon contamination, amplicon sequencing, biodiversity, ribosomal RNA

## Abstract

Arbuscular mycorrhizal fungi (AMF) have been shown to reduce plant stress and improve their health and growth, making them important components of the plant-root associated microbiome, especially in stressful conditions such as petroleum hydrocarbons (PHs) contaminated environments. Purposely manipulating the root-associated AMF assemblages in order to improve plant health and modulate their interaction with the rhizosphere microbes could lead to increased agricultural crop yields and phytoremediation performance by the host plant and its root-associated microbiota. In this study, we tested whether repeated inoculations with a Proteobacteria consortium influenced plant productivity and the AMF assemblages associated with the root and rhizosphere of four plant species growing either in non-contaminated natural soil or in sediments contaminated with petroleum hydrocarbons. A mesocosm experiment was performed in a randomized complete block design in four blocks with two factors: (1) substrate contamination (contaminated or not contaminated), and (2) inoculation (or not) with a bacterial consortium composed of ten isolates of Proteobacteria. Plants were grown in a greenhouse over four months, after which the effect of treatments on plant biomass and petroleum hydrocarbon concentrations in the substrate were determined. MiSeq amplicon sequencing, targeting the 18S rRNA gene, was used to assess AMF community structures in the roots and rhizosphere of plants growing in both contaminated and non-contaminated substrates. We also investigated the contribution of plant identity and biotope (plant roots and rhizospheric soil) in shaping the associated AMF assemblages. Our results showed that while inoculation caused a significant shift in AMF communities, the substrate contamination had a much stronger influence on their structure, followed by the biotope and plant identity to a lesser extent. Moreover, inoculation significantly increased plant biomass production and was associated with a decreased petroleum hydrocarbons dissipation in the contaminated soil. The outcome of this study provides knowledge on the factors influencing the diversity and community structure of AMF associated with indigenous plants following repeated inoculation of a bacterial consortium. It highlights the dominance of soil chemical properties, such as petroleum hydrocarbon presence, over biotic factors and inputs, such as plant species and microbial inoculations, in determining the plant-associated arbuscular mycorrhizal fungi communities.

## 1. Introduction

Plant-root associated microbes have been identified as major contributors to plant survival and health [1], as they can increase growth through a myriad of functions like the secretion of compounds such as phytohormones, organic acids, and antibiotics, thus improving access to nutrition and resistance to pathogens [2,3,4,5]. Moreover, in polluted environments, microorganisms can alleviate soil toxicity by reducing contaminant levels and/or bioavailability [6,7]. Considering the benefits and services microbes provide, their proper management is seen as a way to improve plant health, agricultural yields, as well as the rate of microbe mediated bioremediation. The application of microbial inoculants to the soil in order to modify certain processes and functions is a practice that has grown in popularity over the past few years [8,9]. While results obtained through function screening and cultivation in laboratory conditions are promising, the large-scale application in the field presents challenges in the establishment, persistence, and performance of microbes [10]. Influencing the composition and structure of the rhizosphere-associated microbial community by inoculating indigenous or allochthonous microorganisms is a promising research avenue. It has the potential to be applied at large scale because it does not necessarily rely on the persistence of the inoculants, so long as they induce lasting and desirable changes in microbiome. This could be accomplished through a modification in the soil environment, for example a pH change or nitrogen fixation [11,12], or possibly from an interaction with the native rhizosphere microbes. This would lead to significant changes in the microbial community and plant growth [13,14].

Arbuscular mycorrhizal fungi (AMF) are major actors in the rhizosphere. They are ubiquitous soil microorganisms that develop obligatory associations with the roots of most terrestrial plant species and play an important role in their biological functioning [15]. They provide many benefits to their host plants, notably expanding the plant root system through their extensive mycelial network, which improves plant access to water and mineral nutrition [16,17,18]. Additionally, they can buffer abiotic stresses (i.e., salinity, drought, and trace metal toxicity) [19,20,21] and reduce the damages of root pathogens [22,23,24]. Beyond their interaction with their hosts, AMF mycelia also contribute to soil stability and structure [25] through the secretion of glomalin [26], a long-lasting glycoprotein known for increasing soil aggregation and water retention [27]. Previous experiments showed that plant rhizosphere inoculation with AMF improved the reduction of petroleum contaminants in comparison to non-inoculated plants [28,29]. As there is no information about the ability of AMF to degrade organic pollutants due to their lack of the required enzymes to break them down, the improved degradation of hydrocarbons is likely due to an AMF-mediated increase in plant roots exudation and/or stimulation of other microbial taxa. These taxa can then carry on the degradation of organic pollutants [30,31,32]. Some studies have shown the specific attachment to AMF hyphae of certain types of bacteria isolated from AMF spores and agricultural soils [33,34], and the potential of these microbes to contribute to AMF P uptake [35]. Very few AMF taxa are commercially available for large-scale use [36]. Moreover, the success of AMF inoculation in polluted sites relies on many parameters including the soil and pollutant characteristics and the ability of the inoculated strains to compete with indigenous taxa. Therefore, in an effort to better understand how to influence plant-associated AMF communities, we investigated the contribution of plant species identity and repeated bacterial inoculation in shaping AMF assemblages in contaminated and uncontaminated substrates. We selected plants which were spontaneously growing in highly PH-contaminated sediments, as this would increase their chances of survival. We measured plant growth and PH dissipation at the end of the experiment. This inoculation-based approach could lead to a better understanding of the recruitment and management of the most widespread plant–fungus symbioses on earth in the service of phytotechnology.

## 2. Materials and Methods

Sediments contaminated with petroleum hydrocarbons were collected in October 2013 from a by-product sedimentation basin of a petrochemical plant located at Varennes, on the south shore of the St. Lawrence River near Montreal, Quebec, Canada (45°41′56” N, 73°25′43” W). This basin was artificially made to collect wastewater of the petrochemical plant. It was used for several decades. The petrochemical plant stopped its industrial activities in 2008, and the site was colonized by spontaneous vegetation. No information on the vegetation nor arbuscular mycorrhizal fungal diversity were available for the site before its exploitation. Basic chemical characteristics of the sediments have been previously described [14,37]. Contaminated sediments were collected from the 0–10 cm layer of the decantation basin and brought back to the laboratory where they were thoroughly homogenized and transferred into 60 × 29 × 12 cm trays to a final volume of 18 L per tray and used as the growth substrate for plants. Non-contaminated field soil was also collected from a land plot adjacent to the contaminated basins and transferred to similar trays. Chemical characteristics of both substrates are available in Appendix A. Initial hydrocarbons concentrations were 3055 ± 188 mg/kg for C10–C50 and 35.4 ± 2.6 mg/kg for polycyclic aromatic hydrocarbons (PAHs) in the contaminated treatments. Soils with these levels are considered to be significantly contaminated for agricultural, residential, and commercial use according to the Canadian soil quality guidelines.

### 2.1. Seeds Collection and Germination

We collected seeds from eight species of annual plants that spontaneously grew within the same contaminated basin in October 2013. Out of these eight species, we chose the following four plant species: *Persicaria lapathifolia* (L.) Delarbre, *Lythrum salicaria* L., *Lycopus europaeus* L., and *Panicum capillare* L., based on the availability of a sufficient number of seed pods on the day of collection, as well as on their germination success. The seeds were stratified in damp sand at 4 °C during 8 weeks, after which they were germinated in a 1:1 (*v*/*v*) sand/calcined montmorillonite clay (Turface, Buffalo Grove, IL, USA) mix incubated at room temperature. Germinated seeds were planted in 50 mL multi-cell compartments filled with double autoclaved at 121 °C all-purpose commercial potting soil mix (Scotts Premium potting soil, Rocky View County, AB, Canada). After three weeks of growth, seedlings were carefully transferred to the trays.

### 2.2. Media Preparation and Isolation of Bacteria from the Contaminated Sediments

One kilogram of contaminated sediments was mixed with one liter of sterile demineralized water and stirred for 24 h at room temperature. Subsequently, the supernatant was filtered through a 40 μm sieve and used to prepare a culture medium containing solely 8 g/L of agarose. After being autoclaved for 30 min at 121 °C and left to cool for 45 min, the medium was supplemented with 100 mg/L cycloheximide to inhibit any fungal growth. An inoculum was made up by serially diluting down to 10^−7^ a thoroughly vortexed stock suspension composed of 1 g of the same contaminated sediment in 9 mL of sterile demineralized water. From these serial dilutions, aliquots of 100 μL from dilutions of 10^−6^ and 10^−7^ were spread on the culture plates, incubated at 27 °C for one week, and checked daily for bacterial growth. Each colony was subcultured on the same medium in order to obtain a pure culture. The bacterial isolates were then stored at 4 °C. The same procedure was repeated using an impoverished tryptic soy agar medium (3 g/L) to isolate more copiotrophic bacteria.

### 2.3. 16S rDNA Amplification, Sequencing, and Identification of Bacteria

Bacterial isolates were individually picked with 1 μL sterile inoculation loops and directly added to a PCR master mix. 16S rRNA sequences were amplified using ‘27f’ 5′AGAGTTTGATCMTGGCTCAG 3′ and ‘1492r’ 5′ TACGGYTACCTTGTTACGACTT 3′ primers. The PCR master mix was made up of 1× PCR buffer, 0.5 mg BSA, 2mM MgCL_2_, 0.2 μM of each primer, 0.2 mM of each deoxynucleotide triphosphate, one unit of *Taq* DNA Polymerase (Qiagen, Toronto, ON, Canada), and 1 μL of bacterial cells as DNA template. Thermal cycling conditions were as follows: initial denaturation at 94 °C for 3 min; 30 cycles at 95 °C for 30 s, 55 °C for 30 s, and 72 °C for 1 min; and a final elongation step at 72 °C for 10 min. PCR reactions were performed using an Eppendorf Mastercycler ProS (Eppendorf, Mississauga, ON, Canada). Sanger DNA sequencing was achieved using a commercial service provided at the Genome Quebec Innovation Centre (Montreal, QC, Canada). Briefly, a polymerase chain reaction (PCR) was first conducted using Big Dye 3.1 terminators and thermocyclers. The sequence was then determined by capillary electrophoresis, then analyzed using the ABI 3730xl Data Analyzer (Thermo Fisher Scientific, Mississauga, ON, Canada). Obtained sequences were identified using the NCBI nucleotide BLAST database. Identity was assigned based on the closest match with the highest coverage of our query sequences.

### 2.4. Selection of Isolates and Production of a Bacterial Consortium

A bacterial consortium that was used for the inoculation of plants was formed from a selection of bacterial isolates obtained from the contaminated sediments, as described above. To prepare this consortium, ten bacterial isolates belonging to the phylum Proteobacteria were selected (Appendix A). Protobacteria were chosen because they have been identified as a dominant bacterial group within the same contaminated sediments in previous studies [38,39]. Selected bacterial isolates were individually subcultured in impoverished Tryptic Soy Broth (TSB, 3 g/ L) at 27 °C for 72 h, after which the liquid cultures were centrifuged at 5000 rpm for 10 min at 4 °C. The supernatant was then discarded, and the resulting bacterial cell pellet was re-suspended in 1 L of a sterile isotonic 0.154 M NaCl solution. Cell counts for each isolate were then performed using a Neubauer improved hemocytometer (Sigma-Aldrich, Oakville, ON, Canada), and the inoculum was made up by suspending all the selected isolates in equal numbers in a final volume of 2.4 L at a final concentration of 2.4 × 10^9^ CFU/mL.

### 2.5. Mesocosm Experiment Design and Inoculation

A randomized complete block design with two factors—substrate contamination and inoculation—with four replicates each was setup in four blocks (one replicate per block). The treatments were the following: non-contaminated field soil not inoculated (NB-), non-contaminated field soil inoculated with the bacterial consortium (NB+), contaminated sediments not inoculated (CB-), and contaminated sediments inoculated with the bacterial consortium (CB+). Each tray was planted with the plant species listed above in three rows, separated by 7.5 cm. Within each row, the four plant species were randomly distributed in four planting points, each composed of a cluster of four individual seedlings from the same plant species. These points were spaced at 12.5 cm intervals. Plants were watered several times weekly, as needed throughout the experiment. No fertilization was applied. Bacterial inoculation was performed twice, two and four weeks after planting the trays. At both applications, each tray received 300 mL of the bacterial consortium, which gave a concentration of 4 × 10^7^ CFU/g of dry soil. The planting design in each tray as well as pictures of the experimental setup are available in the Appendix A.

### 2.6. Data Collection and Harvest

The plants were harvested after 16 weeks of growth. At harvest, each plant cluster was carefully removed from the substrate to avoid root damage, and aerial parts were separated from the roots. The substrate was gently shaken off from the roots, and the portion still attached (considered as the rhizospheric soil) was collected and used for DNA extraction. Subsequently, the root system was thoroughly washed with tap water to eliminate the remaining substrate particles. In total, there were 12 rhizospheric soil samples and 12 root samples from each tray (one sample per plant cluster, i.e., three samples for each plant species), hence 384 samples were taken for the whole experiment. All samples were stored at −80 °C until processing. Aerial plant parts were dried in the oven for 72 h at 60 °C and weighted. A plant sampling example can be found in Appendix A. In addition, soil composite samples (3 subsamples) from each of the contaminated mesocosms were collected for petroleum hydrocarbon analysis. C10–C50 aliphatic hydrocarbons and polycyclic aromatic hydrocarbons levels analysis was performed using a commercial service (AGAT labs, Montreal, QC, Canada).

### 2.7. DNA Extraction, PCR Amplification and Illumina MiSeq Sequencing

Total genomic DNA was extracted from plant roots and rhizospheric soil samples using the Nucleospin^®^ Soil Kit (Macherey Nagel, Bethlehem, PA, USA) following the manufacturer’s instructions. The isolated DNA was then diluted tenfold in Milli-Q type 1 sterile water to reduce the risk of PCR inhibition by PH contaminants and humic substances. A two-step nested PCR was performed in order to amplify part of the 18S rRNA gene of arbuscular mycorrhizal fungi using primers AML1 (5′-ATCAACTTTCGATGGTAGGATAGA-3′) and AML2 (5′-GAACCCAAACACTTTGGTTTCC-3′) according to Lee et al. [40] for the first step, followed by an in-house set of internal primers: nu-SSU-0595-5’ CGGTAATTCCAGCTCCAATAG / nu-SSU-0948-3’ TTGATTAATGAAAACATCCTTGGC with overhang adapter sequences to produce a final amplicon size of ~400 bp [41]. Finally, Illumina Miseq-specific indexes were attached to the generated amplicons using the Nextera XT V2 kit using a limited-cycles PCR as recommended by the manufacturer. The indexed sequences were then purified and normalized using the SequalPrep™ Normalization Plate Kit (Thermo Fisher Scientific, St-Laurent, QC, Canada), after which they were pooled at equimolar concentration and sequenced on an Illumina MiSeq sequencer. The 384 samples were multiplexed in one flow cell for sequencing using the 600 cycle MiSeq Reagent Kit v.3 in 2×300 bp configuration (Illumina Inc., San Diego, CA, USA).

### 2.8. Evaluation of the AMF Colonization of Plants Roots

Composite root samples from each plant cluster were surface rinsed in tap water to remove any remaining soil debris and cut in fragments 1 cm long. They were then cleared in a 10% KOH solution, acidified using acetic acid, and stained with trypan blue, following a microwave-assisted protocol [42]. Root fragments were mounted onto microscope slides in lactoglycerol medium (300 mL lactic acid, 300 mL glycerol, 400 mL double-distilled water), and colonization percentages were evaluated using the magnified intersections method [43], with 150 intersects examined per plant cluster for the presence of mycorrhizal structures.

### 2.9. Sequence Processing and Details of the Pipeline

Read assembly and primer trimming were done in Mothur (v.1.34.4). The rest of the processing was performed in QIIME (v.1.9), following the Brazilian Microbiome Project 18S profiling pipeline [44,45]. In Mothur, the reads from each sample were assembled using the ‘make.contigs’ command. This generated a .fasta file containing the assembled reads. Primers were then removed with ‘trim.seqs’, after which the sequences were exported to QIIME. Special labels were then added for compatibility, with the ‘add.qiime.labels.py’ command. Using the Usearch7 sequence analysis tool via QIIME implementation, the dataset was reduced to unique sequences using ‘-derep_fulllength’, which we sorted by decreasing cluster size and removed singletons with ‘-sortbysize’. The sequences were then clustered by operational taxonomic unit (OTU) using a 97% identity threshold using the UPARSE method, and further sequencing errors were removed with ‘– uchime_denovo’. Representative sequences of the OTUs were aligned with ‘-align.seqs’ using the Silva eukaryote database (release 132), and filtered (‘-filter.alignment’ -e 0.10 and –g 0.80). We then produced an OTU table at 97% similarity, and sequences that did not classify as Glomeromycota were removed with ‘filter_taxa_from_otu_table.py’. Finally, we subsampled the sequences, so each sample had the same number of sequences (150). Raw sequence data have been deposited in NCBI’s Sequence Read Archive and can be found under BioProject number PRJNA591301. An operational taxonomic unit (OTU) table of the dataset used for the analyses can be found in Appendix A.

### 2.10. Statistical Analyses

The effect of inoculation with the bacterial consortium on plant biomass and concentration of C10–C50 petroleum hydrocarbons and polycyclic aromatic hydrocarbons (PAH) at the end of the experiment were analyzed using ANOVA in JMP^®^ 11.0.0 statistical software (SAS Institute Inc.). The effect of inoculation and plant species identity on the alpha diversity indices of AMF communities (Chao1, Shannon, and Pielou’s equitability) was also analyzed with ANOVA. The effects of the inoculation with the bacterial consortium and plant species identity on AMF community structure in the rhizosphere were analyzed using R (v3.2.0, The R Foundation for Statistical computing). Principal coordinate analysis (PCoA) based on Bray–Curtis dissimilarity matrixes was used to reveal the effect of treatments and biotope (plant roots and rhizospheric soil) on AMF communities’ structures, while Permanova was used to test the effect of treatments on the beta diversity of AMF communities in the substrate.

### 2.11. Phylogenetic Analysis

Sequences of the operational taxonomic units (OTUs) obtained from the bioinformatic pipeline were combined with all the AMF sequences from Kruger et al. [46], along with the closest matches from the MaarjAM database [47], and aligned using the MUSCLE v.3.8 [48] plugin in Geneious v.6 [49]. We determined the DNA substitution model in jModelTest2 v.2.1.9 [50,51] using the Bayesian information criterion calculations. A phylogenetic analysis was then performed using the Mr.Bayes v.3.2.6 [52] plugin in Geneious v.8 (Biomatters, Auckland, New Zealand). We adjusted the temperature parameter for heating chains in order to keep the swap acceptance rate between 10% and 70%. The number of trees saved was 6000, and the first 1000 trees were ignored before the computation of the consensus tree with the Bayesian posterior probabilities. The tree was rooted using outgroup sequences.

## 3. Results

### 3.1. Sequence Processing 

Sequencing of 18S rRNA gene amplicons generated a total of 13,087,745 reads, of which 5,873,061 usable AMF sequences with an average length of 408 bp were retained after processing and quality filtering. All non-glomeromycotan sequences were excluded from the dataset. Reads number per sample ranged from 20 to 42,410. The dataset was subsampled to 150 sequences per sample for all analyses, in order to account for the disparity in reads numbers while minimizing the loss of samples. Sixteen samples out of 384 did not meet this cutoff criterion and were ignored for the analyses. Only OTUs with a cumulative abundance greater than 0.5% for the whole dataset were considered, yielding 23 OTUs at a 97% similarity threshold. Good’s coverage indices ranged between 96% and 100% for all samples, indicating that most of the diversity was captured.

### 3.2. Diversity and Identity of AMF Taxa

All alpha diversity results can be found in Table 1. AMF diversity was significantly influenced by inoculation, soil contamination, plant species identity, and biotope, as shown by the Shannon diversity indices of AMF taxa (Appendix A). There were also significant inoculation*contamination, plant species*biotope, and biotope*contamination interactions. Overall, AMF communities in the non-contaminated soil had significantly higher Shannon diversity indices than those in the contaminated soil. Inoculation was also significantly associated with higher scores, but this effect was only observed in the non-contaminated soil. Moreover, significantly different diversity scores between plant species were only observed in the roots and not in the rhizosphere: *P. lapathifolia*-associated AMF communities in the roots had a significantly lower Shannon diversity score than those associated with the other plant species (Appendix A). Chao1 species richness estimator was significantly influenced by substrate contamination, biotope, and plant species (Appendix A). There was significant plant species*inoculation, plant species*biotope, and plant species*contamination interactions. Overall, Chao1 estimated species richness was significantly higher in the non-contaminated soil and in the roots in comparison to the contaminated sediments and rhizosphere respectively (Appendix A). The estimated richness was differentially influenced by inoculation among plant species, biotope, and contamination as indicated by the significant interaction terms. There was no significant difference between plant species that received inoculation, whereas richness was significantly higher in *L. salicaria* than *P. capillare* and *P. lapathifolia* non-inoculated treatments. Additionally, estimated richness was similar between plant species in the non-contaminated soil, but significantly higher in *L. salicaria* than *P. capillare* and *P. lapathifolia* (Appendix A). Pielou’s equitability index reflects how evenly species (in this case OTU) are represented in each sample. It varies between 0 and 1, with 1 meaning that all OTUs in the samples are represented by the same number of sequences [53]. We found that AMF communities’ evenness was significantly influenced by the substrate contamination, inoculation, and biotope (Appendix A). Overall, inoculated AMF assemblages showed a significantly higher evenness than those from the non-inoculated communities. Additionally, there was significant plant species*biotope and biotope*contamination interactions. Evenness was not different among plant species in the rhizosphere, but there was significant difference in the roots, with *L. europaeus* showing higher scores than *P. lapathifolia*. Moreover, evenness was similar in the roots and rhizosphere of plants in the contaminated soil, but it was significantly higher in the roots of plants in the non-contaminated soil in comparison with the rhizosphere (Appendix A). The 23 AMF OTUs belonged to 15 virtual taxa (VTX) based on the MaarjAM’s database (Öpik et al., 2010). The taxonomic affiliation of each OTU was confirmed by the phylogenetic analysis (Appendix A). The OTUs belonged to Glomeraceae (16), Claroideoglomeraceae (4), Acaulosporaceae (1)*,* Diversisporaceae (1), and Paraglomeraceae (1) families. Overall, OTUs belonging to Glomeraceae and Claroideoglomeraceae accounted for most of the sequences, irrespective of inoculation, plant species, biotope (roots, rhizosphere), and contamination. They represented more than 96% of the sequences in the contaminated soil and more than 73% in the non-contaminated. In the roots of plants growing in non-contaminated soil, sequences belonging to *Glomus* sp. were the most abundant irrespective of inoculation. Relative abundances ranged from 27.8% (VTX00247) to 40.6% (VTX00143) in the contaminated mesocosms (Figure 1A). On the other hand, rhizospheric samples were dominated by *Paraglomus* sp. VTX00281 with values ranging between 39.4% and 41.6% (Figure 1A). In the non-contaminated sediments (Figure 1B), root samples were dominated by an OTU identified as VTX00114, which belong to *Rhizophagus irregularis* and ranged from 55.8% to 63%. The rhizosphere was dominated by *Claroideoglomus* VTX00193, representing 54.8% to 57.6% of the AMF community.

### 3.3. AMF Community Structure 

Substrate contamination influenced the AMF communities, as illustrated by the PCoA analysis of all samples showing that the samples from contaminated and non-contaminated soil mostly clustered closely together on opposite sides of the biplot (Figure 2A). Permanova analysis showed a significant effect of the substrate contamination level and biotope (Appendix A). Noteworthy is the effect of contamination that explained 52.6% of the variation in the AMF communities (*p* = 0.001). There was also a significant contamination*biotope interaction. In each substrate contamination level, the plant biotope (roots vs. rhizosphere) significantly influenced the structure of AMF assemblages, as shown by the PCoA analysis, with a cleaner separation between communities in the non-contaminated soil (Figure 2B,C). Moreover, UPGMA (Unweighted pair group method with arithmetic mean) hierarchical clustering analysis on the average relative abundance of each of the AMF OTUs in the experimental blocks also showed the influence of biotope on AMF assemblages, as roots and rhizosphere samples belonged to different clusters (Figure 3A,B). The biotope explained 3% of the variation in the AMF communities (*p* = 0.001) in the non-contaminated substrate, and 5.7% in the contaminated sediments (*p* = 0.001), as per the Permanova analysis (Appendix A). In the non-contaminated substrate, AMF communities clustered together in the rhizosphere based on the inoculation, and to a lesser extent in the roots following UPGMA clustering analysis (Figure 3B). On the other hand, inoculation did not induce a clear clustering in the contaminated soil, especially in the roots (Figure 3A). PCoA analysis showed that the root and rhizosphere AMF communities from the non-contaminated substrate underwent a shift in structure following inoculation (Figure 2B), as confirmed by the Permanova with inoculation explaining 3.4% of the variation (*p* = 0.003) in the roots, while in the rhizosphere, inoculation influenced 7.2% of the community structure variation (*p* = 0.001). On the other hand, inoculation significantly influenced the AMF communities only in the roots from the contaminated substrate and accounted for 2% of the variation (*p* = 0.028) and did not have a significant influence in the rhizosphere (Figure 2C). Since the interaction between substrate contamination level and plant biotope was significant, we also assessed the influence of plant species identity on the AMF community structure within each substrate contamination level and biotope using Permanova. In the non-contaminated substrate, plant species identity accounted for 9.5% (*p* = 0.001) of the variation in the roots and 6.7% in the rhizosphere (*p* = 0.003). Similarly, in the contaminated substrate, plant identity significantly explained 9.7% (*p* ≤ 0.001) of the variance in the roots and 5.7% (*p* = 0.009) in the rhizosphere. More details on the Permanova results can be found in Appendix A.

### 3.4. Plant Dry Biomass

Substrate contamination significantly affected plant biomass since plants growing in the contaminated substrate produced significantly less dry biomass than those growing in the non-contaminated substrate for all plant species (*p* < 0.0001). Plant species identity also showed a significant effect (*p* < 0001), as well as inoculation (*p* = 0.0043) (Figure 4A,B). There was a significant contamination * plant species effect (*p* < 0001), where biomass production was significantly different between plant species in the non-contaminated soil, but not in the contaminated setting (Appendix A).

### 3.5. AMF Root Colonization 

The microscope examination of trypan blue-stained roots revealed the occurrence of AMF structures associated with the roots, such as hyphopodia, intraradical mycelium, arbuscules, and vesicles (Appendix A). Assessment of the percentage of root length colonized using the magnified intersect method showed that the colonization rate was significantly higher (*p* < 0.001) in the roots of plants that grew in the contaminated sediments than those in non-contaminated soil for all plant species (Appendix A). Colonization ranged from 25% to 67% of root length in the contaminated sediments and from 12% to 23.5% in the non-contaminated soil (Figure 5A). Inoculation did not influence root colonization in the non-contaminated soil, but it significantly increased the root colonization in the contaminated sediments (Figure 5A).

### 3.6. Effect of Inoculation on PH Concentrations

Bacterial inoculation also influenced the concentrations of PAHs (Figure 5B) and aliphatic hydrocarbons (C10–C50 fraction; Figure 5C) in the contaminated sediments at the end of the experiment. Intriguingly, there was significantly less PAHs (*p* = 0.0181) in the non-inoculated compared to the inoculated treatments (Figure 5B). Values ranged from 6 ± 1.14 mg/kg in the non-inoculated to 10.7 ± 1.11 mg/kg in the inoculated microcosms. The same trend was observed for aliphatic petroleum hydrocarbons (Figure 5C), with levels ranging from 1360 ± 147 mg/kg in the non-inoculated to 2050 ± 502 mg/kg in the inoculated microcosms.

## 4. Discussion

The present study shows that the physicochemical properties of the plant substrate that was contaminated or not with petroleum hydrocarbons strongly shaped the AMF communities associated with the plant roots and rhizosphere. Moreover, within a given substrate contamination level, plant identity, biotope, and inoculation with the bacterial consortium also had significant effects on the structure of AMF assemblages.

### 4.1. Contamination and Biotope Shape AMF Communities

In this experiment, AMF rhizospheric and root communities of plants growing in non-contaminated soil and PH-contaminated sediments were inoculated using a bacterial consortium formed of multiple Proteobacteria isolates. Overall, neither plant species identity nor bacterial inoculation were significant determinants of AMF root and rhizosphere assemblages, obviously because the effect of soil contamination and the differences between biotopes were so large that it overwhelmed the more subtle effects of the other factors, which still were enlightened by interactions such as inoculation*biotope in the non-contaminated soil, inoculation*plant species in the rhizosphere, and contamination*inoculation in both roots and rhizosphere. However, looking at each level of soil contamination, we found significant effect of biotope (rhizosphere vs. roots) and, to a lesser extent but also significant, plant species identity and bacterial inoculation effects. It is still unclear how to manipulate different environmental factors in order to influence the AMF communities to a predictable state, as multiple studies found different patterns. For example, Xu et al. [54] sampled *Chenopodium ambrosioides* plants from five different sites and found, using high-throughput sequencing, that soil properties were the major factor explaining the variation in AMF communities, followed by soil habitat (rhizosphere vs. roots), while plants showed little effect [54]. Contrarily, Krüger et al. [55] found that it was the plant communities that shaped the AMF assemblages, rather than soil properties, during primary succession on mine spoils [55]. In another study, the authors analyzed rhizospheric soil from different plants sampled from five sites in the Tibetan alpine steppe; they found that plant species identity did not significantly explain the variation in the AMF assemblages, but rather it was the precipitation which was associated with an increased hyphal length density [56]. Finally, Dumbrell et al. [57] sampled 28 plants species growing along a pH gradient and found that it was the pH, rather than plants, that structured AMF communities [57]. While of different conclusions, the aforementioned studies showed that the soil environment is generally at the center of the process of shaping the AMF communities; therefore, an approach where the soil parameters would be precisely manipulated might be a successful method to shape microbial assemblages. Such shaping of microbial communities has been observed in the human gut microbiome, where significant, reproducible, and long-lasting changes were induced by osmotic stress caused by the laxative polyethylene glycol, which disrupts the mucus barrier and causes an IgG response from the immune system against the resident highly abundant bacteria and a modification of the cytokinin levels [58]. This showed that a modification of the environmental parameters triggered a cascade of reactions that led to lasting and predictable changes. In the present experiment, we tried to modify the community structure through a biotic disturbance in the form of a repeated inoculation of a bacterial consortium. We hypothesized that this inoculation could change the soil environment (at least temporarily) through the bacterial metabolites produced by the members of the consortium and their interaction with resident microorganism, as well as the decomposition of dead bacterial cells. As a result, the AMF community structure would be altered since AMF have been shown to interact with their surrounding microorganisms [59,60,61,62]. In fact, the bacterial inoculation did cause significant shifts in the AMF community structure in both biotopes (roots vs. rhizosphere) growing in non-contaminated field soil, but in the contaminated sediments only the AMF community of roots exhibited significant shifts. The toxicity caused by the high concentrations of PH might likely make the growing environment more selective for stress-tolerant strains of AMF, thus overriding the influence of the biotic disturbance. On the other hand, while insignificant in the rhizosphere, inoculation caused significant shifts in the AMF in roots, suggesting a possible plant-mediated restructuring of the assemblages. This further complicates the task of producing a reproducible and predictable method for shaping community structure, as it was shown that plant species identity counts in the selection of microbial associates.

### 4.2. Glomeraceae and Claroideoglomeraceae Dominate Most Samples

OTUs belonging to the Glomeraceae dominated the roots of all plant species in both contaminated and non-contaminated substrates, and with or without inoculation. OTUs related to *Glomus* sp. were the most abundant in the roots of plants in non-contaminated substrate, while in the contaminated substrate roots were dominated by *Rhizophagus* sp. VTX114 in. On the other hand, an OTU identified as *Paraglomus* sp. dominated the rhizosphere soil of plants in non-contaminated soil, while *Claroideoglomus* sp. was the most abundant in the rhizospheric soil in contaminated sediments. Previous field studies in the same geographical area from which the substrates used in this experiment were sourced have reported higher abundances of *Paraglomus* in non-contaminated soil and *Claroideoglomus* in contaminated sediments, as well as a dominance of OTUs related to *Rhizophagus* sp. in plant roots in comparison to rhizospheric soil [60,63,64]. We found that the structure of AMF in the roots of plants was different at each level of contamination; however, the communities were dominated by Glomeraceae strains in both cases, while the rhizosphere was dominated by non-Glomeraceae OTUs. AMF colonize roots using different strategies; for example, some species produce a larger mycelial network in the rhizosphere before penetrating the roots (e.g., *Gigaspora* spp.), while others readily colonize the roots without producing an extensive network of mycelium outside the roots (e.g., *Glomus* spp.) [65]. This could be explained by the fact that root colonization strategies of AMF species vary with their taxonomic affiliation, and that members of the Glomeraceae family more readily colonize the roots than produce extraradical structures [66]. Such species would be ideal candidates for inoculating plants in contaminated environments to assure adequate plant colonization.

### 4.3. Contamination Increases Root AM Colonization

We found that AMF root colonization percentages in the four plant species collected from the contaminated sediments were significantly higher than their counterpart grown in non-contaminated soil. De la Providencia, Stefani, Labridy, St-Arnaud, and Hijri [63] sampled during two successive years *P. capillare* plants from the same location from which the contaminated sediments for the present experiment originated. They found that in *P. capillare*, colonization percentages were 4.84% in 2011 and 12% in 2012 [63], which were lower than the levels found in the same plant species in this experiment. Plants are subjected to the natural elements and different edaphic factors in situ, which might explain the observed difference. It should also be noted that contamination levels found at the site were much higher than those recorded in this greenhouse experiment, reaching up to 41,000 mg/kg of C10–C50 hydrocarbons. This difference likely was due to the natural weathering process, microbial activity, and volatilization. Higher levels of PHs might stimulate a stronger mycorrhizal colonization of the roots as observed here; nonetheless, extreme levels are likely to inhibit it. Cabello [67] found that the AMF root colonization in plants sampled from two contaminated sites in Argentina and Germany was lower than in the same plant species sampled from non-contaminated sites [67]. On the other hand, colonization percentages of over 80% were recorded in plants growing in weathered crude oil in the Ecuador Amazon region [68]. Could the PH-induced stress have driven the plants into engaging more symbioses and promoted the growth of the AMF, which in turn could help alleviate the toxic effects? During the colonization of the roots, AMF develops multiple structures such as arbuscules and vesicles, which play an important part in the interaction with the host plant and the nutrition of the fungus [69]. Defoliation such as grazing by herbivores induces plant stress, in part due to loss of nutrients present in the foliage [70]. In a manual defoliation experiment, *Medicago truncatula* plants showed significantly lower vesicular colonization percentages than their non-defoliated counterparts [71]. Piippo et al. [72] found that grazing simulation through defoliation of two varieties of the biennial grassland herb *Gentianella amarella* decreased arbuscular colonization in the early flowering type, but on the other hand increased it in late flowering type plants [72]. Saito et al. [73] also observed a decrease of root colonization following the defoliation of grazing intolerant grass *Myscanthus sinensis* and at one sampling point for the grazing tolerant *Zoysia japonica*. Inversely, Ambrosino et al. [74] noted little to no effect of defoliation on total root AMF colonization. These contradictory results suggest that plant/host responses to stress can vary from one plant species to the other, and that different stress types would induce different responses, but also point to the host plant identity as a potential key for influencing AMF root communities. The bacterial inoculation in this experiment was associated with significantly increased root colonization percentages, but only in the contaminated substrate. AMF activity through its extra radical mycelium has been found to be influenced by soil microbiota assemblages [75]. The stress-inducing toxicity of PH contaminants could explain why significant changes in root colonization were only observed in the contaminated substrate, as it might have led to more dynamic and responsive root/AMF interactions.

### 4.4. Inoculation Affects Plant Growth and PH Attenuation

In this experiment, bacterial inoculation was associated with increased plant growth for some of the species; nevertheless, this did not translate to an increased PH attenuation rate. On the contrary, it significantly decreased the degradation rate, resulting in greater concentrations in the inoculated compared to the non-inoculated trays at the end of the experiment.

One would assume that the health of plants is likely to be a key component of the success or failure of the process. Nonetheless, our results suggest that improved plant growth does not necessarily lead to a successful remediation of PH contaminants. Wu et al. [76] used dual inoculation of ryegrass with a PAH-degrading bacteria and an AMF strain in order to attenuate PAHs levels, and they found that this approach was superior to single inoculation for the degradation of pyrene and phenanthrene. However, inoculation with the bacteria *Acinetobacter* sp. did not significantly increase ryegrass growth. Moreover, Bell et al. [77] showed that specialized soil bacterial assemblages obtained in culture are less efficient at degrading crude oil than a diverse microbiome.

## 5. Conclusions

In light of these results, it seems that a multifaceted approach should be favored in order to successfully shape the rhizosphere microbiome structure and function. It should take into account the physicochemical parameters of the soil environment, with a selection of plant/microbial assemblages that can persist in the given setting, while leading to predictable and reproducible effects on the plant host and soil.

## Figures and Tables

**Figure 1 microorganisms-08-00602-f001:**
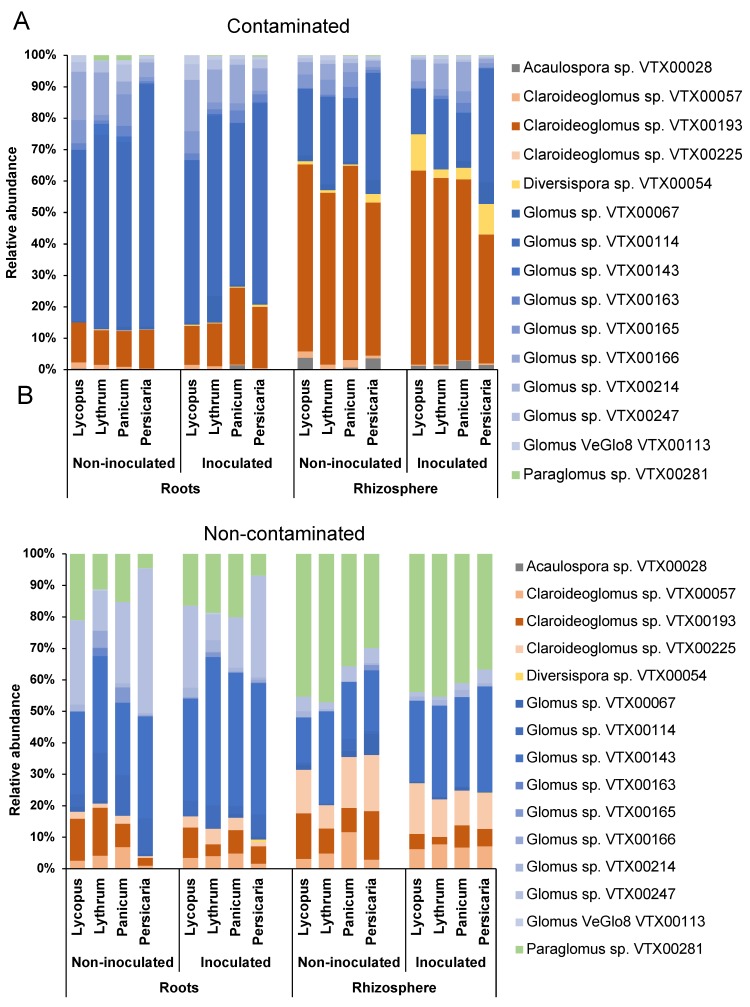
Relative abundance of AMF taxa detected in the roots and rhizospheres of plants from the contaminated substrate (**A**) and the non-contaminated substrate (**B**)**,** inoculated or not-inoculated with the bacterial consortium. LE: *Lycopus europaeus*; LS: *Lythrum salicaria*; PC: *Panicum capillare*; PL: *Persicaria lapathifolia*.

**Figure 2 microorganisms-08-00602-f002:**
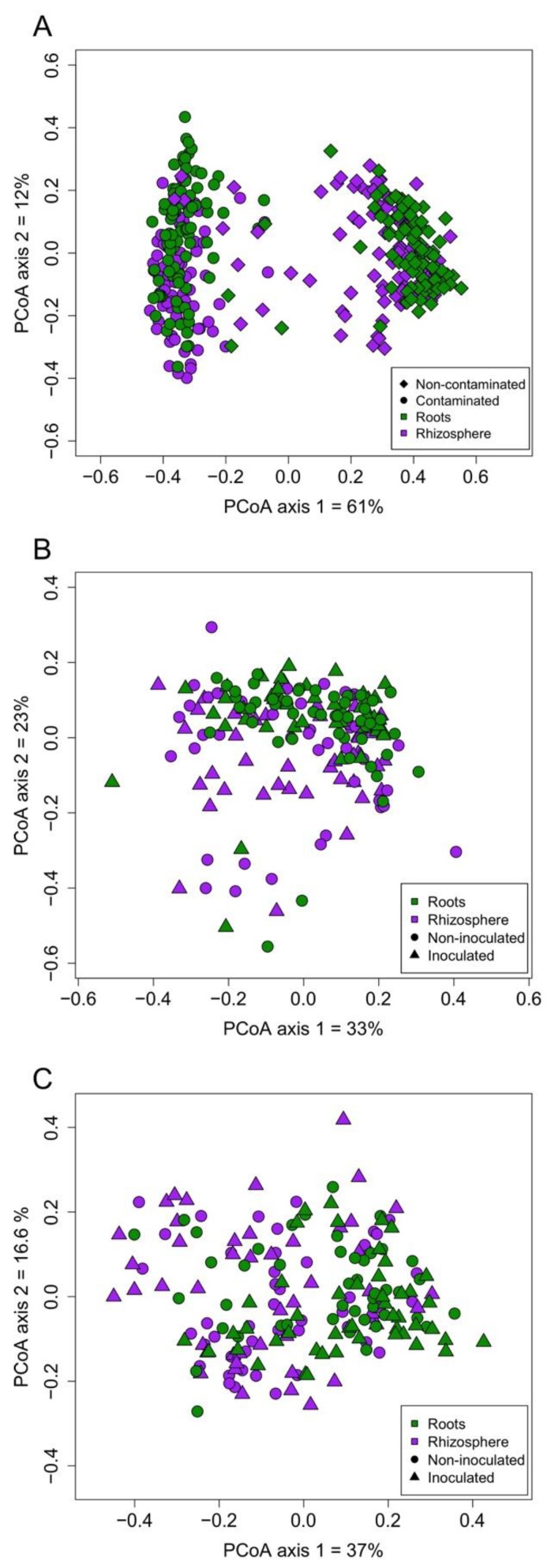
Principal coordinate analysis based on the Bray–Curtis dissimilarity of (**A**) roots and rhizosphere AMF communities together from both type of substrates (non-contaminated soil or contaminated sediments), and separately from (**B**) the non-contaminated soil and (**C**) the contaminated sediments.

**Figure 3 microorganisms-08-00602-f003:**
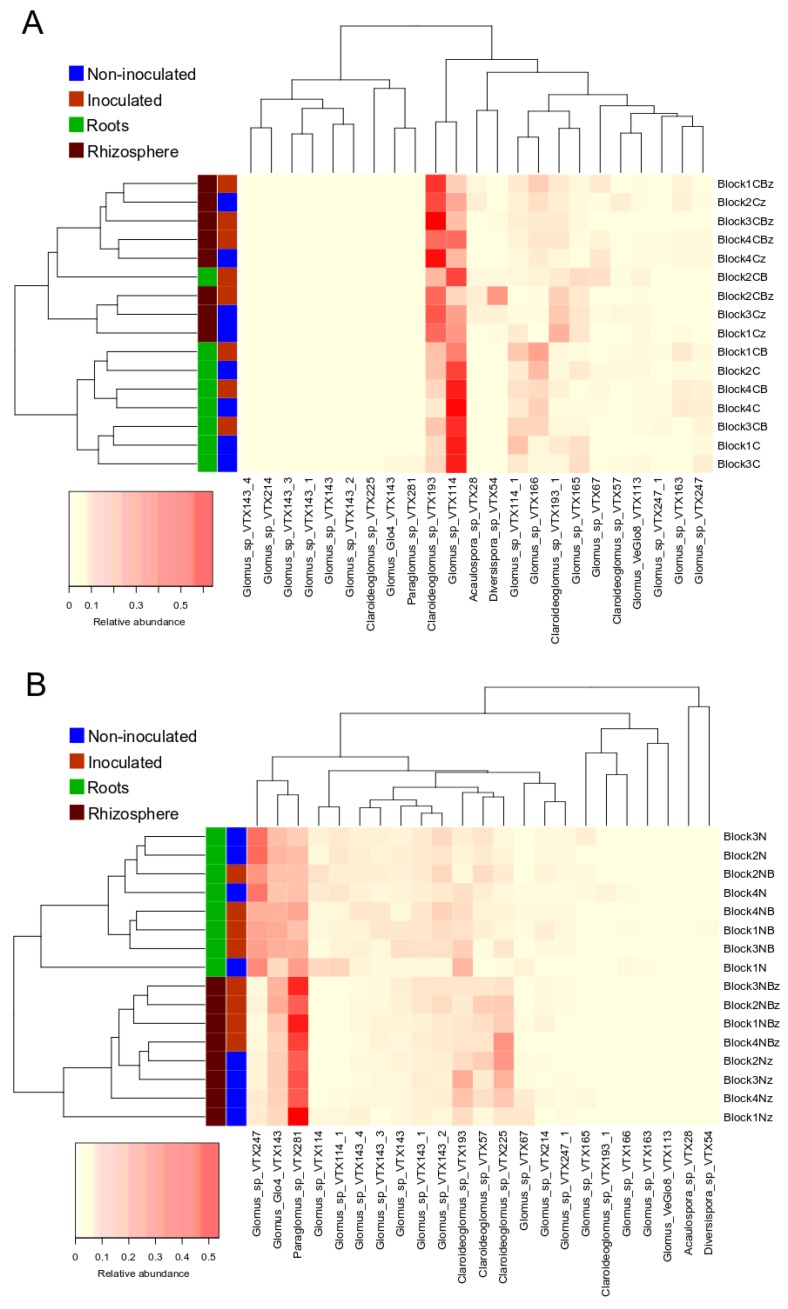
Heat map and UPGMA clustering based on the relative abundance of AMF OTUs in roots and rhizosphere in each block, from the different treatments in the contaminated soil (**A**) and the non-contaminated sediments (**B**). The colors of the last two squares on the left of rows represent the origin and treatment of each sample (biotope and inoculation).

**Figure 4 microorganisms-08-00602-f004:**
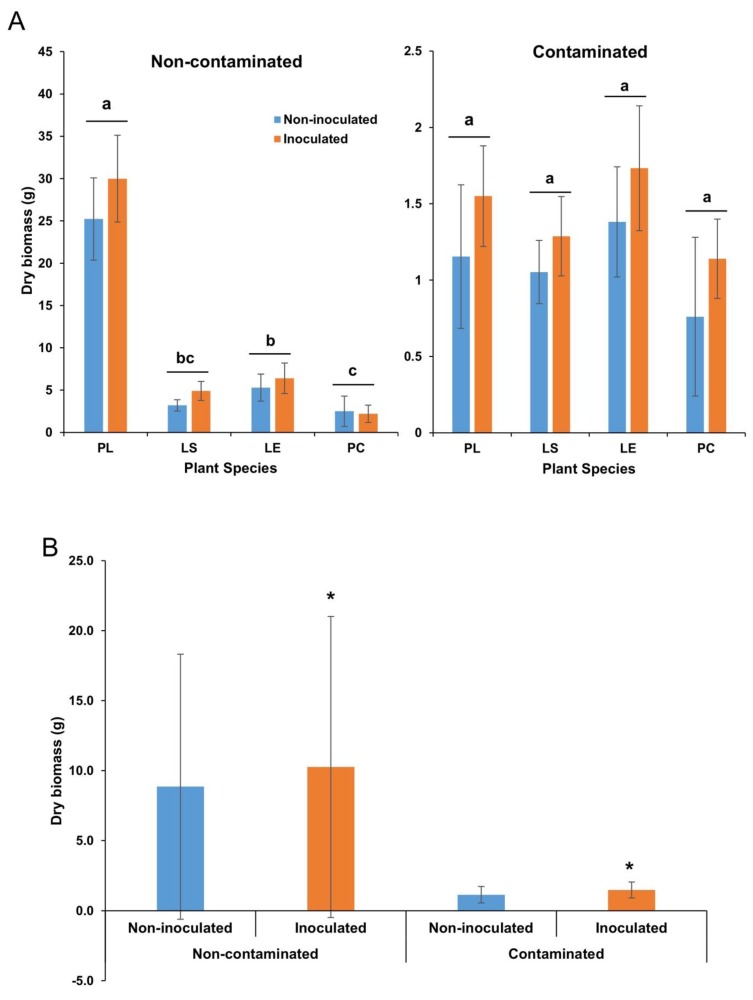
Effect of inoculation on average plant aerial dry biomass for each plant species (**A**) and overall (**B**), in the non-contaminated soil and the contaminated sediments. Errors bars are standard deviations. Within each contamination level, treatments not sharing the same letter are significantly different (**A**). Asterisks indicate a significant difference between the two treatments in each contamination level (**B**). PL: *Persicaria lapathifolia*; LS: *Lythrum salicaria*; LE: *Lycopus europaeus*; PC: *Panicum capillare*. *N* = 12 for each species.

**Figure 5 microorganisms-08-00602-f005:**
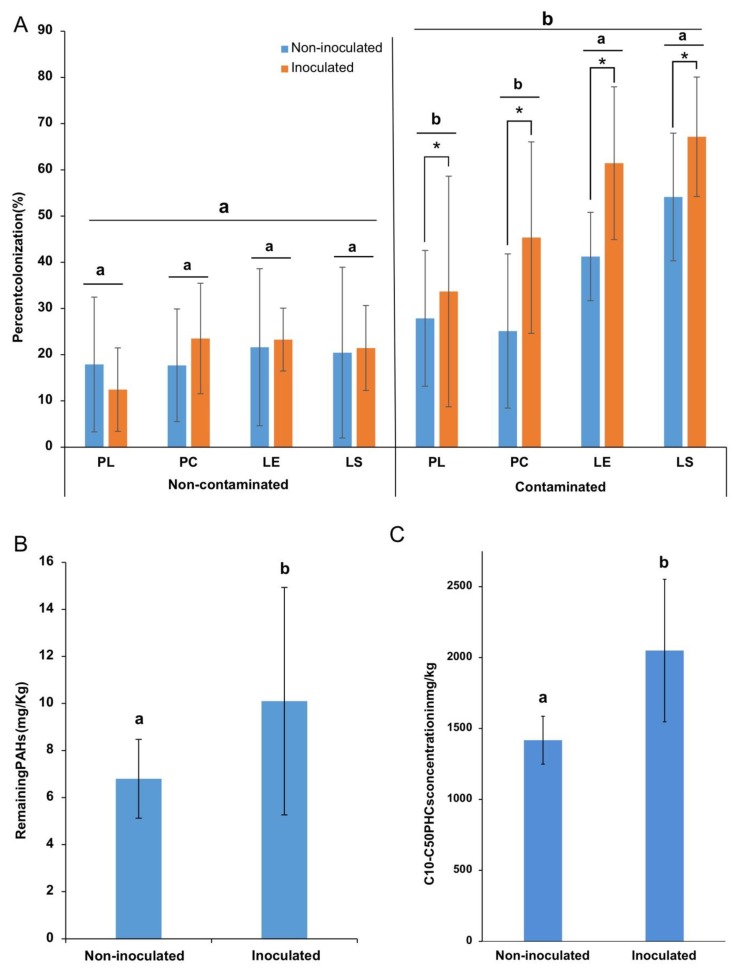
(**A**) Average AMF root colonization percentage for each plant species in both soil contamination levels. PL: *Persicaria lapathifolia*; LS: *Lythrum salicaria*; LE: *Lycopus europaeus*; PC: *Panicum capillare***.** Within each contamination level, plant species not sharing the same letter are significantly different. For each plant species, an asterisk denotes a significant difference between inoculation treatments. (**B**) Average remaining PAHs and (**C**) C10/C50 PHs in the contaminated sediments at harvest. Errors bars are standard deviations.

**Table 1 microorganisms-08-00602-t001:** Alpha diversity indices of AMF communities as calculated in the different treatments and biotopes.

				Shannon		Chao 1		Pielou’s Equitability	
Contamination	Inoculation	Biotope	Plant Species	Mean	Std Dev	Mean	Std Dev	Mean	Std Dev
**Contaminated**	**Inoculated**	**Rhizosphere**	**LE**	1.7	0.38	9.96	3.35	0.56	0.12
			**LS**	1.96	0.43	8.92	2.37	0.64	0.14
			**PC**	1.94	0.43	10.88	4.43	0.62	0.13
			**PL**	1.61	0.39	8.6	3.57	0.57	0.13
		**Roots**	**LE**	2.39	0.22	10.87	1.38	0.71	0.07
			**LS**	1.87	0.64	10.32	3.67	0.57	0.16
			**PC**	2.08	0.27	8.58	1.43	0.69	0.07
			**PL**	1.73	0.5	8.42	3.05	0.58	0.11
	**Non-inoculated**	**Rhizosphere**	**LE**	1.86	0.36	9.3	2.82	0.62	0.11
			**LS**	1.87	0.5	7.74	2.16	0.65	0.14
			**PC**	1.74	0.65	8.86	4.5	0.61	0.14
			**PL**	1.68	0.32	7.7	2.63	0.62	0.12
		**Roots**	**LE**	2.09	0.27	9.25	1.78	0.67	0.07
			**LS**	2.25	0.42	17.5	2.93	0.6	0.09
			**PC**	1.75	0.62	8.58	2.85	0.59	0.15
			**PL**	1.39	0.51	8.36	3.71	0.48	0.13
**Non-contaminated**	**Inoculated**	**Rhizosphere**	**LE**	2.51	0.28	13.52	4.42	0.71	0.07
			**LS**	2.49	0.25	12	2.5	0.71	0.04
			**PC**	2.6	0.34	12.68	2.08	0.73	0.07
			**PL**	2.79	0.31	15.3	5.96	0.76	0.06
		**Roots**	**LE**	2.92	0.67	14.94	2.87	0.76	0.15
			**LS**	3.05	0.22	14.43	1.65	0.8	0.06
			**PC**	2.82	0.34	14.24	3.34	0.77	0.08
			**PL**	2.79	0.52	13.49	1.95	0.75	0.12
	**Non-inoculated**	**Rhizosphere**	**LE**	2.33	0.37	14.87	4.16	0.65	0.09
			**LS**	2.43	0.33	12.27	1.69	0.68	0.08
			**PC**	2.28	0.44	10.69	2.55	0.68	0.09
			**PL**	2.46	0.57	14.24	3.1	0.67	0.12
		**Roots**	**LE**	2.7	0.93	14	5.29	0.73	0.23
			**LS**	2.82	0.46	14.18	2.82	0.77	0.11
			**PC**	2.59	0.47	12.95	2.43	0.73	0.11
			**PL**	2.47	0.31	13.06	1.85	0.69	0.07

LE: Lycopus europaeus; LS: Lythrum salicaria; PL: Persicaria lapathifolia; PC: Panicum capillare. Std. Dev: Standard deviation.

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
