# Peer review of "Arbuscular Mycorrhizal Fungal Assemblages Significantly Shifted upon Bacterial Inoculation in Non-Contaminated and Petroleum-Contaminated Environments"

_microorganisms, 2020, doi:10.3390/microorganisms8040602_

Round 1

Reviewer 1 Report

Dear Publisher and Authors

The biodiversity of arbuscular mycorrhizae in the roots and rhizosphere of four pant species, grown 16 weeks in soils contaminated (or not) with hydrocarbons and inoculated (or not) with bacterial was investigated and revealed. Smart work supported by recent techniques permitted to collect a lot of desired scientific information.

Results:

The structure of the mycorrhizae groups found in roots and rhizosphere is different in plants growing in contaminated or non-contaminated soils. The inoculation of bacteria influenced the composition of the mycorrhizae in particular in the uncontaminated soils. The species that dominate in roots or rhizosphere of contaminated soils are different from those of uncontaminated soils.

In contaminated soils, plants showed more colonized by mycorrhizae roots than in corresponding normal soils. Contamination causes these plants to react. Repeated inoculation of bacteria collected in these contaminated soils had a positive effect on the growth of some plant species, but a low influence on the toxicity of the soils.

The article is clear, well written and presented. It is necessary to consult almost all the tables shown in the annex.

As a forest soil ecologist, I would have known something about the causes of contamination and the natural value of contaminated sites. Why were those four species used instead of others? Why four instead of two or 20? Even if I'm sure that already with four of them, there was a lot to do.

I would have preferred to know what the site was like before it was altered with petroleum. I wish I had read the list of species, with the related natural mycorrhizae, collected in a site that is still not over-polluted. How have the species changed with pollution and why?

In this way, one begins to think about how to bring the environment back towards a balance where there are no more hydrocarbons in the soil.

The process that is presented in the article is different: we put some plants in a standard environment (to avoid noises that can distort the interpretation), adding some colonies of bacteria and trying to understand how the species react regardless of what nature (the historical coexistence of species over time) had built up with/for them in the previous millions of years.

Environmental management needs to be rethought, including agriculture and life in general. A project that science is called upon to support and coordinate. Can we stop and think about it a little more before continuing?

The usual clash between reductionists and holists. If we are happy enough with the results that the biological sciences have given so far (how are humans and the environment?), there is no need to discuss.

Author Response

We thank the reviewer#1 for taking time to read our manuscript and providing us with comment. We have addressed all the questions point by point below:

As a forest soil ecologist, I would have known something about the causes of contamination and the natural value of contaminated sites. Why were those four species used instead of others? Why four instead of two or 20? Even if I'm sure that already with four of them, there was a lot to do.

  • The rational of selecting these plant species was to use spontaneous vegetation of the site. Therefore, we collected seeds during the fall from 8 plants species out of 23 plant species that were previously identified on the site. However, after stratification of seeds, only 4 plant species we able to germinate and provide enough quantities in order to setup the experiment. This explains the use of 4 plant species.

I would have preferred to know what the site was like before it was altered with petroleum. I wish I had read the list of species, with the related natural mycorrhizae, collected in a site that is still not over-polluted. How have the species changed with pollution and why?

  • The site of study was an artificial basin that was made in the fifties and it was used to collect and store waste water of petrochemical plant. We don’t have any data on the original state of the site. Even, if we had information of top-soil, it wouldn’t represent the initial state of the AMF community because the site has been dug approximately 2 m deep. We had access to that site six years after the plant has stopped its petrochemical activities. During this period, the site was colonized by spontaneous vegetation. We don’t have any information of AMF diversity of the site before our study.

In this way, one begins to think about how to bring the environment back towards a balance where there are no more hydrocarbons in the soil.

The process that is presented in the article is different: we put some plants in a standard environment (to avoid noises that can distort the interpretation), adding some colonies of bacteria and trying to understand how the species react regardless of what nature (the historical coexistence of species over time) had built up with/for them in the previous millions of years.

Environmental management needs to be rethought, including agriculture and life in general. A project that science is called upon to support and coordinate. Can we stop and think about it a little more before continuing?

The usual clash between reductionists and holists. If we are happy enough with the results that the biological sciences have given so far (how are humans and the environment?), there is no need to discuss.

  • It is difficult to address this question. As explained above, the site is man-made. Our study tested whether microbial assisted-phytoremediation could improve petroleum hydrocarbon pollutant degradation.

Reviewer 2 Report

The authors of this study tested the effects of PH contamination, plant species identity and inoculation with native Proteobacteria on AMF communities in roots and rhizosphere of four different plant species as well as their AMF colonization , plant growth and PH attenuation.

I liked the overall concept of the study and enjoyed reading the manuscript in general. However, reversed contamination assignments in Figure 1 created confusion downstream. Please correct and adjust any discussion.

Title:

okay

Abstract:

35-38: Please make the concluding sentence more specific. It seems to too general and does not really highlight your actual findings.

Keywords:

okay

Introduction:

I enjoyed reading the introduction!! 

Material and methods

98-100:

PAH is mentioned first time. Please add 'polycyclic aromatic hydrocarbons'.

Maybe you can make a statement about the severity of the contamination. Is it severe, moderate or mild/light. This will help the general understanding.

122:

can you be more specific? I would recommend to rephrase 'another subset of bacterial colonies' to e.g.  'more oligotrophic bacteria' or whatever was the target community.

124:

typo, correct to : '2.3. 16SrDNA'

You do not mention anything about the identification, although it is in the header. Please provide information about how you identified the bacteria ( database used etc.)

171:

'were' instead of 'was'

218:

This is the first mentioning of 'biotope', please indicate that it stands for roots and rhizosphere, as you did  in L 278.

236:

Please clarify why you chose 150 sequences- could it not have been more?

238-239:

I am not sure if I understand correctly - you subsampled 150 sequences- which were assigned to OTUs ( 97 % threshold). Then you considered only OTUs which had a cumulative abundance of 0.5%, which yielded 23 OTUs. For all 23 OTUs taxonomic identity was assigned at a 97 similarity. Is that what you meant?

Results:

282-284:

In this section of the results, the figures need to be corrected, because the assignment of contaminated and non-contaminated was reversed.

282: needs to be Figure 1B

283: needs to be Figure 1B

284: needs to be Figure 1A

310-312:

In this section of the results, the figures need to be corrected, because the assignment of contaminated and non-contaminated was reversed. I believe

310: needs to be Figure 3B

212: needs to be Figure 3A

333: Incorrect assignment: Figure 3A is contaminated

334: Incorrect assignment: Figure 3B is non-contaminated

366: graph assignment is reversed< B should be C and vice versa

Discussion

389: name specific factors instead of calling them Permanova factors

406: I suggest a transition into the sentence about the Human gut microbiome. E.g ' Such shaping of microbial communities has been observed in the Human gut microbiome, where significant ......'

428-437:

Paragraph needs to be adjusted to corrected Figures. 

430: needs to be 'non-contaminated'

432: needs to be 'contaminated'

433: needs to be 'non-contaminated'

451-456: 

Checking reference 63, it seems that the previous contamination levels were higher than the levels in the current study. You might want to consider the change in contamination levels in your discussion. 

460: rephrase: '...induced stress have driven the plants.....'

463: you might want to consider a transition that you mention defoliation  as a representation of stress.

Graphs:

In general, I thought the graphs were well executed. Nevertheless, I have a some suggestions for correction and improvement.

Figure1:

headers/titles of graphs are reversed;  1A should be non-contaminated and 1B contaminated. It is stated correctly in the legend.

Because you discuss about the taxa in this figure at the genus level, I suggest to color the taxa by genus (e.g. Glomus in various yellow shades or Claroideoglomus in blue shades etc.). That will improve  the general readability of the graph.

Figure3:

Only 22 OTUs are included in these graphs. Why was OTU 327049662 ( Diversispora , VTX 00054) not included?

Looking at the Paraglomus sp.(VTX00281= OTU356658167), there seems to be some disagreement between Figure 1 and Figure 3.  Other taxa ( e.g Rhizophagus sp. (VTX00114 = OTU 569889297) seems to agree in Figure 3 B, but not in Figure3A. Please double-check the data. 

When comparing Figure 1 and Figure 3, it was cumbersome to have to use Table S3 to figure out the taxonomic assignment of each OTU. I suggest to replace the OTU names with the assigned taxonomic names or use a combination.

Figure S3:

graph is hard to read. Maybe landscape orientation and bigger font will help.

Tables:

Table S3:

The numbering in the sample code was confusing.Which one is the block? Is the other number the replication? Please clarify.

Author Response

We thank this anonymous reviewer for providing us with the opportunity to present a more concise and clearer paper. In view of the comments, we have now addressed all comments. I have given the specific details in the Response to the comments of the reviewer of how the manuscript has been changed or how we have addressed the reviewer’s comments.

Abstract:

35-38: Please make the concluding sentence more specific. It seems to too general and does not really highlight your actual findings.

  • We added the following as the concluding sentence: ‘It highlights the dominance of soil chemical properties such as petroleum hydrocarbon presence over biotic factors and inputs, such as plant species and microbial inoculations in determining the plant-associated arbuscular mycorrhizal fungi communities.’

Keywords:

okay

Introduction:

I enjoyed reading the introduction!!

  • Thank you

Material and methods

98-100: PAH is mentioned first time. Please add 'polycyclic aromatic hydrocarbons'.

  • We added the mention 'polycyclic aromatic hydrocarbons

Maybe you can make a statement about the severity of the contamination. Is it severe, moderate or mild/light. This will help the general understanding.

  • We added: Soils with these levels are considered to be significantly contaminated for agricultural, residential and commercial use according to the Canadian soil quality guidelines.

122:

can you be more specific? I would recommend to rephrase 'another subset of bacterial colonies' to e.g.  'more oligotrophic bacteria' or whatever was the target community.

  • We added “to isolate more copiotrophic bacteria”

124:

typo, correct to : '2.3. 16SrDNA'

  • Corrected

You do not mention anything about the identification, although it is in the header. Please provide information about how you identified the bacteria ( database used etc.)

  • We added on line 138: ‘Obtained sequences were identified using the NCBI nucleotide BLAST database. Identity was assigned based on the closest match with the highest coverage of our query sequences.’

171:

'were' instead of 'was'

  • Changed

218:

This is the first mentioning of 'biotope', please indicate that it stands for roots and rhizosphere, as you did  in L 278.

  • We added ‘(plant roots and rhizospheric soil)’ following the mention of biotope

236:

Please clarify why you chose 150 sequences- could it not have been more?

  • We added: The data set was subsampled to 150 sequences per sample for all analyses, in order to account for the disparity in reads numbers while minimizing the loss of samples

238-239:

I am not sure if I understand correctly - you subsampled 150 sequences- which were assigned to OTUs ( 97 % threshold). Then you considered only OTUs which had a cumulative abundance of 0.5%, which yielded 23 OTUs. For all 23 OTUs taxonomic identity was assigned at a 97 similarity. Is that what you meant?

  • Actually the whole dataset was first assigned to OTUs with a 97% threshold, after which we excluded OTUs with less than 0.5% cumulative abundance and ended up with 23 OTUs. Taxonomic affiliation was then assigned through the phylogenetic analysis at 97%, and finally we subsampled at 150 sequences for the analyses. 

Results:

282-284:

In this section of the results, the figures need to be corrected, because the assignment of contaminated and non-contaminated was reversed.

282: needs to be Figure 1B

283: needs to be Figure 1B

284: needs to be Figure 1A

  • The figure orders was correct, it was the legends that were reversed, therefore we corrected the legends to show Figure 1A as contaminated and 1B as non-contaminated

310-312:

In this section of the results, the figures need to be corrected, because the assignment of contaminated and non-contaminated was reversed. I believe

310: needs to be Figure 3B

212: needs to be Figure 3A

333: Incorrect assignment: Figure 3A is contaminated

334: Incorrect assignment: Figure 3B is non-contaminated

  • Here also we corrected the legend and reversed Figure 3A as contaminated and Figure 3B as non-contaminated

366: graph assignment is reversed< B should be C and vice versa

  • We corrected the legend and assigned Figure 5B for PAHs and Figure 5C for C10-C50 PHCs

Discussion

389: name specific factors instead of calling them Permanova factors

  • We modified the sentence to include them as following: ‘….. which still were enlightened by interactions such as inoculation*biotope in the non-contaminated soil, inoculation*plant species in the rhizosphere and contamination*inoculation in both roots and rhizosphere’

406: I suggest a transition into the sentence about the Human gut microbiome. E.g ' Such shaping of microbial communities has been observed in the Human gut microbiome, where significant ......'

  • We modified the beginning of the sentence to the following: ‘Such shaping of microbial communities has been observed in the human gut microbiome, where significant, reproducible, and long lasting changes were induced……’

428-437:

Paragraph needs to be adjusted to corrected Figures. 

430: needs to be 'non-contaminated'

432: needs to be 'contaminated'

433: needs to be 'non-contaminated'

  • The modifications of the figures’ legends mentioned above makes their mentions in the paragraph correct.

451-456: 

Checking reference 63, it seems that the previous contamination levels were higher than the levels in the current study. You might want to consider the change in contamination levels in your discussion. 

  • At line 456 we added: ‘It should also be noted that contamination levels found at the site were much higher than those recorded in this greenhouse experiment, reaching up to 41000 mg/kg of C10-C50 hydrocarbons. This difference was likely due to the natural weathering process, microbial activity, and volatilization. Higher levels of PHs might stimulate a stronger mycorrhizal colonization of the roots as observed here nonetheless, extreme levels are likely to inhibit it.’

460: rephrase: '...induced stress have driven the plants.....'

  • We rephrased as suggested by the reviewer.

463: you might want to consider a transition that you mention defoliation as a representation of stress.

  • We added a transition as follows: Defoliation such as grazing by herbivores induces plant stress, in part due to loss of nutrients present in the foliage [70]. In a manual defoliation experiment, Medicago truncatula plants showed……

Graphs:

In general, I thought the graphs were well executed. Nevertheless, I have a some suggestions for correction and improvement.

Figure1:

headers/titles of graphs are reversed;  1A should be non-contaminated and 1B contaminated. It is stated correctly in the legend.

  • We corrected the legend so that 1A is contaminated and 1B is non-contaminated

Because you discuss about the taxa in this figure at the genus level, I suggest to color the taxa by genus (e.g. Glomus in various yellow shades or Claroideoglomus in blue shades etc.). That will improve  the general readability of the graph.

  • We changed the colors of the taxa to different shades of blue and orange.

Figure3:

Only 22 OTUs are included in these graphs. Why was OTU 327049662 ( Diversispora , VTX 00054) not included?

  • We redid the analysis to include the missing OTU.

Looking at the Paraglomus sp.(VTX00281= OTU356658167), there seems to be some disagreement between Figure 1 and Figure 3.  Other taxa ( e.g Rhizophagus sp. (VTX00114 = OTU 569889297) seems to agree in Figure 3 B, but not in Figure3A. Please double-check the data. 

  • We redid the analysis from the raw data and plotted the heatmaps. They now reflect the same patterns as in figure 1. Note that in figure 1, we grouped similar OTUs under 1 VTX (ex: VTX143 was represented by more than 1 OTU. Same with VTX114 and VTX243, etc…), while in the heatmap representation, we showed each OTU individually.

When comparing Figure 1 and Figure 3, it was cumbersome to have to use Table S3 to figure out the taxonomic assignment of each OTU. I suggest to replace the OTU names with the assigned taxonomic names or use a combination.

  • We replaced the OTU numbers by their assigned taxonomy (VTX).

Figure S3:

Graph is hard to read. Maybe landscape orientation and bigger font will help.

  • Improved

Tables:

Table S3:

The numbering in the sample code was confusing.Which one is the block? Is the other number the replication? Please clarify.

  • We modified and clarified the legend.